# Association of maternal obesity with fetal and neonatal death: Evidence from South and South-East Asian countries

**Rezwanul Haque**[1]*, **Syed Afroz Keramat**[2], **Syed Mahbubur Rahman**[3], **Maimun Ur Rashid Mustafa**[4], **Khorshed Alam**[5,6]

1 Faculty of Arts and Social Sciences, Department of Economics, American International University-Bangladesh, Kuratoli, Khilkhet, Dhaka, Bangladesh, 2 Economics Discipline, Social Science School, Khulna University, Khulna, Bangladesh, 3 Faculty of Business Administration, American International University-Bangladesh, Kuratoli, Khilkhet, Dhaka, Bangladesh, 4 Department of Economics, American International University-Bangladesh, Kuratoli, Khilkhet, Dhaka, Bangladesh, 5 School of Business, University of Southern Queensland, Toowoomba, Queensland, Australia, 6 Centre for Health Research, University of Southern Queensland, Toowoomba, Queensland, Australia

* rezwanul_05@yahoo.com

**Data Availability Statement:** The data used in this study are third party data from the DHS program (https://www.dhsprogram.com) and can be

## Abstract

### Background

Obesity prevalence is increasing in many countries in the world, including Asia. Maternal obesity is highly associated with fetal and neonatal deaths. This study investigated whether maternal obesity is a risk factor of fetal death (measured in terms of miscarriage and still-birth) and neonatal mortality in South and South-East Asian countries.

### Methods

This cross-sectional study pooled the most recent Demographic and Health Surveys (DHS) from eight South and South-East Asian countries (2014–2018). Multivariate logistic regression was deployed to check the relationships between maternal obesity with fetal and neonatal deaths. Finally, multilevel logistic regression model was employed since the DHS data has a hierarchical structure.

### Results

The pooled logistic regression model illustrated that maternal obesity is associated with higher odds of miscarriage (adjusted odds ratio [aOR]: 1.26, 95% CI: 1.20–1.33) and still-births (aOR: 1.46, 95% CI: 1.27–1.67) after adjustment of confounders. Children of obese mothers were at 1.18 (aOR: 1.18, 95% CI: 1.08–1.28) times greater risk of dying during the early neonatal period than mothers with a healthy weight. However, whether maternal obesity is statistically a significant risk factor for the offspring's late neonatal deaths was not confirmed. The significant association between maternal obesity with miscarriage, stillbirth and early neonatal mortality was further confirmed by multilevel logistic regression results.

accessed following the protocol outlined in the Methods section.

**Funding:** The author(s) received no specific funding for this work.

**Competing interests:** The authors have declared that no competing interests exist.

**Abbreviations:** aOR, Adjusted Odd Ratio; CI, Confidence Interval; DHS, Demographic Health Survey; LMIC, Low- and Middle-Income Country; SDG, Sustainable Development Goal; WHO, World Health Organization.

## Conclusion

Maternal obesity in South and South-East Asian countries is associated with a greater risk of fetal and early neonatal deaths. This finding has substantial public health implications. Strategies to prevent and reduce obesity should be developed before planning pregnancy to reduce the fetal and neonatal death burden. Obese women need to deliver at the institutional facility centre that can offer obstetrics and early neonatal care.

## Introduction

Preventable obesity has almost tripled since 1975, and the World Health Organization (WHO) estimated that 1.9 billion adults were overweight in 2016, of which 650 million were obese [1,2]. In Asia and the Pacific, the prevalence of overweight and obesity has increased by approximately 18% from 1990 to 2013 [3]. Among the South Asian nations, Afghanistan, Bhutan, Maldives and Pakistan showed a higher rate of increase in the prevalence of obesity and overweight (30% and higher); while Malaysia, Singapore and Thailand from Southeast Asia experienced obesity prevalence in more than 30% of the total population in 2013 [3]. Globally, it was estimated that 40% and 15% of women aged 18 years and above were overweight and obese, respectively, in 2016 [2].

Obesity, a global issue that countries are struggling to address [3], is associated with miscarriage and has been identified as a major health concern in low and middle-income countries (LMICs) [4]. Obesity, among females especially, has attracted researchers for many reasons. Primarily, maternal weight, obesity, or reediness impacts fertility among women; secondly, it could lead to fetal and neonatal death [1,4,5].

The neonatal period is said to be the most vulnerable time for a child. In the early 1990s, 38 deaths per 1000 births occurred; in 2019, the rate decreased to 17 per 1000 live births. However, around 6700 neonatal deaths occur daily in the world [6]. Therefore, neonatal death is a concern for policymakers and researchers around the world. Baroni et al. [7] investigated the neonatal mortality rates in Brazil, whereas Liu et al. [8] and Abdul-Mumin et al. [9] investigated the key reasons for neonatal death and preterm birth in China and Ghana. Regional investigations have identified that preterm birth, intrapartum complications and pneumonia (in case of China), and preterm birth complications and birth asphyxia (in case of Ghana) are the leading causes of neonatal deaths [8,9]. On a global scale, the major causes of neonatal death include preterm birth, severe infections and asphyxia [10]. Maternal obesity posed a risk for neonatal deaths during the first two days in Sub-Saharan Africa [11]. Systematic review with meta-analyses has also found that many risks are associated with the mother's Body Mass Index (BMI) [12]. In Columbia, the BMI of a mother is associated with an infant's weight [5]. Maternal obesity poses more than double the risk of stillbirth in Denmark [13]. Multi-country analysis has found that the rates of stillbirth and neonatal mortality in south Asia are almost double the rates in Sub-Saharan Africa [14].

Previous researchers have mostly considered a single outcome variable; in some cases, two variables were considered to examine the association between maternal obesity with other variables of interest, including miscarriage, stillbirth [4] and neonatal death [8,9]. In contrast, this research has taken a holistic approach to compare and contrast how maternal obesity is related to pre and post-birth maternal and neonatal risk factors by including multiple outcome variables together. Accordingly, this research investigates the association between maternal health conditions in terms of weight with miscarriage, stillbirth and neonatal death. The findings of

the study contribute to the existing literature in three ways. First, this research has taken multiple variables together, which is rarely observed in the previous literature. Second, this study shows a varied level of impacts of maternal obesity on the selected parameters. The regional coverage is also an addition. Third, the research findings would provide insights into the possible further amendments needed to initiate and implement nationwide programs targeting obesity, especially for women, which would be helpful for national-level policymakers.

## Methodology

### Data source and settings

The data collected from the Demographic and Health Surveys (DHS) website (https://www.dhsprogram.com) have been used in this study. DHS is a nationally representative cross-sectional household survey typically conducted in a 5-year interval in selected LMICs. The DHS, well known for its high data quality, uses a standardised questionnaire to facilitate comparisons between cross-country.

### Study participants

This research selected and pooled the most recent surveys from eight out of 15 countries of South and South-East Asia available in the DHS database. These countries include Cambodia (2014), East Timur (2014), India (2015–16), Myanmar (2015–16), Nepal (2016), Maldives (2016–17), Bangladesh (2017–18) and Pakistan (2017–18). The remaining seven countries were excluded from the analysis due to publicly inaccessible, inadequate and/or obsolete data. Initially, the sample was composed of 228,111 women of reproductive age who gave birth in the last five years of the survey. From this number, the following were excluded from the study: women who had twin children (3913); women who were currently pregnant (19,452); mother had flagged BMI (197); and mother's BMI information was missing (10,654). Hence, the final sample limits to 193,895 mothers aged 15–49 years old (Fig 1).

### Outcome variables

The endpoint of the study was the measurement of fetal and neonatal death.

### Fetal death

The general outcome was used to derive the fetal death, which included miscarriage and stillbirth depending on the weeks of gestational age. According to the WHO definition, stillbirth refers to a baby born dead after 28 weeks of gestation. However, several developed countries use 20–24 weeks of gestation as the lower limit of stillbirth [15], which presents a problem in differentiating miscarriage and stillbirth based on gestational time. In this study, following an assessment conducted in Latin America [16], miscarriage is defined as a fetal loss on or before 20 weeks of gestation, and stillbirth is the fetal loss after 20 weeks of pregnancy.

### Early and late neonatal death

This study considered the WHO definition of neonatal death, which occurs from birth to 28 days of infant life. Neonatal death can be further classified into early neonatal death (death between 0 and 7 completed days of birth) and late neonatal death (death between 8 and 28 completed days of birth) [17,18].

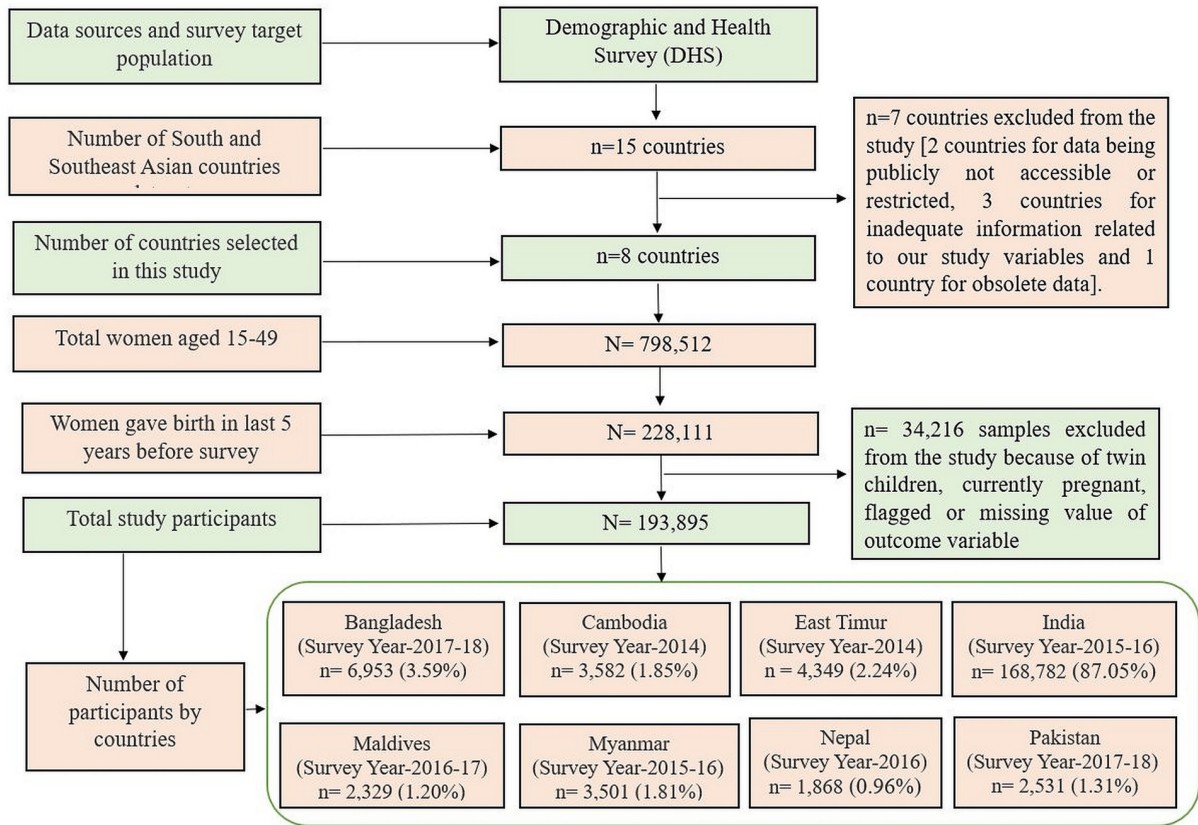

**Fig 1. Distribution of the study participants and year of the survey across eight countries.**

## Exposure variable

Maternal weight status was the exposure of interest, and it was ascertained through the mother's BMI. BMI was calculated as the ratio of weight in kilograms (kg) divided by height in metres-squared ($m^2$), and it was categorised into four groups following the WHO guidelines [2]. Given that the target population of this study was drawn from South and South-East Asia, we used the cut off points of BMI for Asians, which are categorised as: underweight (BMI< 18.50), healthy weight (BMI 18.50 to <23), overweight (BMI 23 to <27) and obese (BMI≥ 27) [19]. The reason for using separate a BMI calculator for Asian and less-developed countries has been discussed in prior studies where the authors considered ethnical diversity that results in small body size and lower gestational weight gain compared to developed countries [20,21].

## Confounders

The study tried to keep most potential confounders after conducting a detailed literature review [11,16,22–24]. Bivariate analysis was conducted, and later, the covariates were included in the fully adjusted model if found significant at 5% or less. The present study attempts to incorporate most of the social, demographic and economic variables used in other studies, such as age, education, employment status, household wealth, civil status, area of residence, age at first marriage, number of children, age at first birth, and place of delivery.

## Estimation strategy

The authors constructed a pooled dataset of eight South and South-Eastern countries and subsequent observations of 193,895 women of reproductive age and their children born in the last five years of the survey. The DHS data involve multistage sampling, unequal sampling probabilities, and stratification. Therefore, before performing any statistical analysis, this study weighted the data using sampling weights, primary sampling unit, strata and cluster to restore the representatives of the survey and futher to calculate standard errors and reliable estimates as per the DHS framework to approximate level-weights [25]. STATA command "svy set" was used for this analysis to address the complex survey design. All analyses such as summary statistics, cross-tabulation and regression were conducted using the STATA software, version 14. To summarise the characteristics of the cohorts, this study first conducted descriptive statistics in the form of frequency (n) and percentage (%). Later, the study reported the frequencies of maternal BMI categories, stillbirth, miscarriage, early neonatal mortality, late neonatal mortality and other covariates. Chi-square tests or t-tests were used to assess the bivariate relationship between all outcome variables with the mother's BMI and other covariates.

This study explored the association between maternal weight status with fetal and neonatal death using the multivariable logistic regression model due to the dichotomous nature of the dependent variables. In the logistic model, the following statistical model is developed to capture the association:

$$Y_i = \alpha_0 + \beta_1 MBMI_i + \beta_2 SD_i + \ldots\ldots\ldots\ldots + \varepsilon_i \tag{1}$$

In Eq 1, $Y_i$ represents the vector of all dependent variables, such as miscarriage, stillbirth, early neonatal death and late neonatal death. $MBMI_i$ is the mother's BMI. Finally, $SD_i$ represents the vector of the socio-economic and demographic characteristics, and $\varepsilon_{it}$ is the error term. To avoid the possible multicollinearity issue, a variance inflation factor test was conducted (not shown). No correlation was found among the explanatory variables. The logistic regression model results were expressed as unadjusted and adjusted Odd Ratio (aOR), and a p-value at <0.05 level was found to be statistically significant.

Since the DHS data has a hierarchical structure, in this study, multivariable multilevel logistic regression analysis was also applied to determine the effects of each independent variable of miscarriage, stillbirth, early neonatal mortality and late neonatal mortality. This study considered two-level multi-level analysis: level 1 and 2 indicated individual level and country level, respectively. The multilevel analysis was conducted separately in two regression models for each outcome variables: miscarriage, stillbirth, early neonatal mortality and late neonatal mortality. Model 1 represented a null model was fitted without independent variables to test random variability in the intercept. Null model also used to estimate the intra-class correlation coefficient (ICC). Model 2, a mixed model, included all independent variables which were found to be statistically significant (<0.05) in the bivariate analysis, was examined the effects of all considered independent variables simultaneously.

This study estimated both fixed and random effect parameters from multivariate multilevel modelling. The fixed and random effect models measured the adjusted odds ratios and country-level variance, respectively. Furthermore, Intraclass correlation coefficient (ICC), calculating the variance in percentage explained by the country-level factors, were used to measure the random effects.

We explored the interactions of exposure variable of mother's BMI with potential confounding variables of age, area of residence, and mother's education for four outcome variables (e.g., miscarriage, stillbirth, early neonatal mortality and late neonatal mortality). However,

no statistically significant associations were found. Therefore, results are not reported in the paper.

## Results

Table 1 summarises miscarriage, stillbirth, early neonatal mortality, late neonatal mortality, mother's BMI and socio-demographic characteristics of the study participants. A total of 193,895 women were included in the analysis. Among the participants, 10.50% reported fetal death, of which 9.14% and 1.36% reported miscarriage and stillbirth, respectively. Early neonatal death was reported by 4.42%, whereas late neonatal death was less than 1%. The descriptive statistics also revealed the prevalence of overweight (19.64%) and obese (9.77%) mothers.

The bivariate relationship between the mother's BMI and each outcome variable, miscarriage, stillbirth, early neonatal mortality, late neonatal mortality and other covariates achieved through the Chi-square tests or t-tests are displayed in Table 2. The bivariate analyses showed that all the outcome variables were significantly associated with the mother's BMI and all the other confounders except employment status at a 5% level of significance. This study incorporated employment status in the multivariate regression model to check whether employment status was associated with outcome variables at any level.

Table 3 presents the pooled estimates of the association between a mother's BMI with fetal and neonatal death. To facilitate interpretation, the present study displayed the results in the form of unadjusted and adjusted odds ratios of outcome variables with a change in the level of mother's BMI. The adjusted model demonstrated that the mother's BMI was a significant predictor of miscarriage, stillbirth and early neonatal death. The adjusted model reveals that the odds of miscarriage among the overweight and obese mothers were 1.08 (aOR:1.08, 95% CI: 1.04–1.13) and 1.26 (aOR: 1.26, 95% CI: 1.20–1.33) times higher, respectively, compared with mothers possessing healthy weight. Mothers with obesity were more likely to report stillbirth (aOR:1.46, 95% CI: 1.27–1.67) than those with healthy BMI ranges. The mothers with obesity had elevated odds (aOR: 1.18, 95% CI: 1.08–1.28) of having early neonatal death. No significant association was found between maternal obesity and late neonatal death.

Table 4 illustrates the results of multivariate multilevel logistic regression analysis for null and country-level factors for measuring the random effect of country and fixed effects of factors associated with miscarriage, stillbirth, early neonatal mortality and late neonatal mortality in eight south and south-east Asian countries.

The null model (Model 1) of miscarriage described that significant variation exists in the odds across the countries ($\tau = 0.72$; 95% CI: 0.44–1.18; p<0.001). The variance estimate was greater than zero, it indicates that there were country differences in miscarriage among women in South and South-East Asian countries, and thus multilevel analysis should be considered as an appropriate approach for further analysis. Likewise, after controlling the effect of independent variables at the country level, the variance at the country level had a significant impact ($\tau = 0.70$; 95% CI: 0.43–1.15; p<0.001) in Model 2. The null model (Model 1) demonstrated that, overall, 13.59% variation in the odds of miscarriage was reported that involved cluster difference of the characteristics (ICC = 13.59%). The variation between clusters declined to 12.92% in Model 2. The result of Model 2 exposed that the adjusted odds of miscarriage among obese mothers were 1.19 (aOR: 1.19; 95% CI: 1.13–1.26) times higher compared to mothers with a healthy weight.

Concerning the second outcome variable, stillbirth, both the null model (Model 1) and mixed model (Model 2) confirmed significant variation (variance estimate greater than zero) in the odds across the country ($\tau = 1.01$; 95% CI: 0.60–1.69; p<0.001 and $\tau = 0.99$; 95% CI: 0.58–1.66; p<0.001 respectively). The variation of odds between the clusters declined from

**Table 1. Weighted background characteristics of the study participants.**

|  | Frequency (n) | Weighted percentage (%) |
|---|---|---|
| **Outcome variable** |  |  |
| Miscarriage |  |  |
| no | 17,768 | 90.86 |
| yes | 176,127 | 9.14 |
| Stillbirth |  |  |
| no | 191,279 | 98.64 |
| yes | 2,616 | 1.36 |
| Early neonatal mortality |  |  |
| no | 185,189 | 95.58 |
| yes | 8,706 | 4.42 |
| Late neonatal mortality |  |  |
| no | 192,406 | 99.25 |
| yes | 1,489 | 0.75 |
| **Exposure variable** |  |  |
| Mothers' BMI |  |  |
| Underweight | 43,540 | 23.41 |
| Healthy weight | 94,913 | 47.18 |
| Overweight | 37,920 | 19.64 |
| Obesity | 17,522 | 9.77 |
| **Confounding variables** |  |  |
| Age |  |  |
| <20 years | 6,754 | 3.78 |
| 20–29 | 123,232 | 66.01 |
| 30–39 | 56,632 | 27.12 |
| 40 and over | 7,277 | 3.09 |
| Education |  |  |
| No formal education | 52,259 | 25.75 |
| Primary | 30,708 | 15.55 |
| Secondary | 89,614 | 46.49 |
| Higher | 21,314 | 12.21 |
| Employment status |  |  |
| No occupation | 140,002 | 73.63 |
| Employed | 53,893 | 26.37 |
| Wealth index |  |  |
| Poorest | 46,225 | 22.71 |
| Poor | 43,680 | 20.96 |
| Middle | 39,073 | 19.95 |
| Richer | 34,537 | 19.29 |
| Richest | 30,380 | 17.10 |
| Civil status |  |  |
| Non-partnered | 4,429 | 1.97 |
| Married | 189,466 | 98.03 |
| Area of residence |  |  |
| Urban | 50,798 | 29.73 |
| Rural | 143,097 | 70.27 |
| Age at first marriage |  |  |
| <18 years | 69,295 | 37.62 |

*(Continued)*

**Table 1.** (Continued)

| | Frequency (n) | Weighted percentage (%) |
|---|---|---|
| *18 or over* | 124,600 | 62.38 |
| Number of children | | |
| *None* | 1,392 | 0.70 |
| *1–3* | 162,610 | 85.78 |
| *4–6* | 26,846 | 12.26 |
| *6+* | 3,047 | 1.26 |
| Age at first birth | | |
| *<18 years* | 26,032 | 14.02 |
| *18–29 years* | 143,678 | 74.60 |
| *30–40* | 19,579 | 9.42 |
| *over 40* | 4,606 | 1.96 |
| Place of delivery | | |
| *Home* | 29,199 | 13.23 |
| *Public health centre* | 83,507 | 40.16 |
| *Private health center* | 41,265 | 25.13 |
| *Others*: NGOs | 39,924 | 21.48 |

23.73% to 22.95% from Model 1 to Model 2 (ICC = 23.73% and ICC = 22.95%, respectively). The result from the mixed-model showed that mothers with obesity were more likely to report stillbirth (aOR:1.31, 95% CI: 1.14–1.51) compared with those with a healthy BMI range.

In the case of early neonatal mortality, variation in the odds across the country was greater than zero which confirms that there were country differences in early neonatal mortality among women in south and south-east Asia. The variance at the country level reduced from null model ($\tau$ = 0.39; 95% CI: 0.23–0.57; p<0.001) to mixed model ($\tau$ = 0.37; 95% CI: 0.21–0.64; p<0.001). The variation of odds between the clusters declined from 4.42% to 3.96% from Model 1 to Model 2. The mixed model results revealed that the mothers with obesity had elevated odds (aOR: 1.20, 95% CI: 1.10–1.31) of having early neonatal death compared with mothers possessing healthy weight. However, no significant association was found in the mixed model between maternal obesity and late neonatal death.

## Discussion

This study examined the association between maternal obesity with fetal and neonatal death. The present study constructed a pooled data of eight South and South-East Asian countries and used multivariable logistic regression for estimation to derive the association. The study also performed multilevel logistic regression to assess the country level variation. In both cases, maternal obesity is associated with an increased risk of miscarriage, stillbirth and early neonatal death.

In this study, maternal obesity was associated with greater odds of miscarriage, which was in line with previous studies conducted in Nepal [4], Brazil [16], London [26], North England [27] and in the United Kingdom [28,29]. A pooled analysis of a systemic review has found a higher miscarriage rate in obese mothers than in mothers with normal BMI [30]. In another systematic analysis, Marchi et al. found an even greater risk of miscarriage in pregnant mothers with obesity than those with optimum weight [31]. Obesity comprises an array of other risk factors that possibly result in preterm pregnancy loss. Early research has shown that obesity increases the risk of neural tube defect and hypertensive disorder of pregnancy, which

**Table 2. Bivariate analysis between maternal obesity with fetal and neonatal death in the south and south-east Asia.**

| | Miscarriage (n = 17,768) | | P value | Stillbirth (n = 2,616) | | P value | Early neonatal mortality (n = 8,706) | | P value | Late neonatal mortality (n = 1,489) | | P value |
|---|---|---|---|---|---|---|---|---|---|---|---|---|
| | n | % (CI) | | n | % (CI) | | n | % (CI) | | n | % (CI) | |
| **Exposure variable** | | | | | | | | | | | | |
| Mothers' BMI | | | <0.001 | | | <0.021 | | | <0.001 | | | <0.016 |
| Underweight | 3,531 | 19.87 (19.29–20.47) | | 596 | 22.78 (21.22–24.43) | | 2,204 | 25.32 (24.41–26.24) | | 380 | 25.52 (23.37–27.80) | |
| Healthy weight | 8,349 | 46.99 (46.26–47.72) | | 1,261 | 48.20 (46.29–50.12) | | 4,226 | 48.54 (47.49–49.59) | | 724 | 48.62 (46.09–51.16) | |
| Overweight | 3,813 | 21.46 (20.86–22.07) | | 481 | 18.39 (16.95–19.92) | | 1,516 | 17.41 (16.63–18.22) | | 262 | 17.60 (15.74–19.61) | |
| Obesity | 2,075 | 11.68 (11.21–12.16) | | 278 | 10.63 (9.50–11.87) | | 760 | 8.73 (8.15–9.34) | | 123 | 8.26 (6.97–9.77) | |
| **Confounding variables** | | | | | | | | | | | | |
| Age | | | <0.001 | | | <0.001 | | | <0.001 | | | <0.001 |
| <20 years | 414 | 2.33 (2.12–2.56) | | 49 | 1.87 (1.42–2.47) | | 201 | 2.31 (2.01–2.65) | | 45 | 3.02 (2.26–4.02) | |
| 20–29 | 11,998 | 67.53 (66.83–68.21) | | 1,773 | 67.78 (65.96–69.54) | | 4,821 | 55.38 (54.33–56.42) | | 794 | 53.32 (50.78–55.85) | |
| 30–39 | 4,866 | 27.39 (26.74–28.05) | | 729 | 27.87 (26.18–29.62) | | 3,242 | 37.24 (36.23–38.26) | | 563 | 37.81 (35.38–40.30) | |
| 40 and over | 490 | 2.76 (2.53–3.01) | | 65 | 2.48 (1.95–3.16) | | 442 | 5.08 (4.64–5.56) | | 87 | 5.84 (4.76–7.15) | |
| Education | | | <0.001 | | | <0.001 | | | <0.001 | | | <0.001 |
| No formal education | 3,800 | 21.39 (20.79–22.00) | | 889 | 33.98 (32.19–35.82) | | 3,349 | 38.47 (37.45–39.49) | | 607 | 40.77 (38.29–43.28) | |
| Primary | 2,920 | 16.43 (15.90–16.99) | | 469 | 17.93 (16.50–19.45) | | 1,678 | 19.27 (18.46–20.12) | | 301 | 20.21 (18.25–22.33) | |
| Secondary | 8,803 | 49.54 (48.81–50.28) | | 1,093 | 41.78 (39.90–43.68) | | 3,253 | 37.37 (36.35–38.39) | | 501 | 33.65 (31.29–36.09) | |
| Higher | 2,245 | 12.64 (12.15–13.13) | | 165 | 6.31 (5.44–7.31) | | 426 | 4.89 (4.46–5.37) | | 80 | 5.37 (4.34–6.64) | |
| Employment status | | | <0.992 | | | <0.796 | | | <0.012 | | | < 0.73 |
| No occupation | 12,830 | 72.21 (71.54–72.86) | | 1,883 | 71.98 (70.23–73.67) | | 6,183 | 71.02 (70.06–71.96) | | 1,081 | 72.6 (70.28–74.81) | |
| Employed | 4,938 | 27.79 (27.14–28.46) | | 733 | 28.02 (26.33–29.77) | | 2,523 | 28.98 (28.04–29.94) | | 408 | 27.4 (25.19–29.72) | |
| Wealth index | | | <0.001 | | | <0.001 | | | <0.001 | | | <0.001 |
| Poorest | 3,208 | 18.05 (17.50–18.63) | | 738 | 28.21 (26.52–29.97) | | 2,852 | 32.76 (31.78–33.75) | | 545 | 36.6 (34.19–39.08) | |
| Poor | 3,793 | 21.35 (20.75–21.96) | | 667 | 25.50 (23.86–27.20) | | 2,283 | 26.22 (25.31–27.16) | | 364 | 24.45 (22.33–26.69) | |
| Middle | 3,808 | 21.43 (20.83–22.04) | | 554 | 21.18 (19.65–22.79) | | 1,670 | 19.18 (18.37–20.02) | | 286 | 19.21 (17.28–21.29) | |
| Richer | 3,551 | 19.99 (19.40–20.58) | | 392 | 14.98 (13.67–16.40) | | 1,179 | 13.54 (12.84–14.28) | | 157 | 10.54 (9.08–12.21) | |
| Richest | 3,408 | 19.18 (18.61–19.77) | | 265 | 10.13 (9.03–11.35) | | 722 | 8.29 (7.73–8.89) | | 137 | 9.20 (7.83–10.78) | |
| Civil status | | | <0.001 | | | <0.038 | | | <0.003 | | | <0.295 |
| Non-partnered | 234 | 1.32 (1.16–1.50) | | 44 | 1.68 (1.25–2.25) | | 159 | 1.83 (1.57–2.13) | | 28 | 1.88 (1.30–2.71) | |
| Married | 17,534 | 98.68 (98.50–98.84) | | 2,572 | 98.32 (97.75–98.75) | | 8,547 | 98.17 (97.87–98.43) | | 1,461 | 98.12 (97.29–98.70) | |
| Area of residence | | | <0.001 | | | <0.013 | | | <0.001 | | | <0.001 |
| Urban | 5,582 | 31.42 (30.74–32.10) | | 630 | 24.08 (22.48–25.76) | | 1,777 | 20.41 (19.58–21.27) | | 298 | 20.01 (18.06–22.12) | |

*(Continued)*

**Table 2.** (Continued)

| | Miscarriage (n = 17,768) | | P value | Stillbirth (n = 2,616) | | P value | Early neonatal mortality (n = 8,706) | | P value | Late neonatal mortality (n = 1,489) | | P value |
|---|---|---|---|---|---|---|---|---|---|---|---|---|
| | n | % (CI) | | n | % (CI) | | n | % (CI) | | n | % (CI) | |
| *Rural* | 12,186 | 68.58 (67.90–69.26) | | 1,986 | 75.92 (74.24–77.52) | | 6,929 | 79.59 (78.73–80.42) | | 1,191 | 79.99 (77.88–81.94) | |
| Age at first marriage | | | <0.001 | | | <0.001 | | | <0.001 | | | <0.001 |
| *<18 years* | 6,084 | 34.24 (33.55–34.94) | | 1,095 | 41.86 (39.98–43.76) | | 4,048 | 46.50 (45.45–47.55) | | 699 | 46.94 (44.42–49.49) | |
| *18 or over* | 11,684 | 65.76 (65.06–66.45) | | 2,616 | 58.14 (56.24–60.02) | | 4,658 | 53.50 (52.45–54.55) | | 790 | 53.06 (50.51–55.58) | |
| Number of children | | | <0.001 | | | <0.001 | | | <0.001 | | | <0.001 |
| *None* | 164 | 0.92 (0.79–1.07) | | 50 | 1.91 (1.45–2.51) | | 776 | 8.91 (8.33–9.53) | | 108 | 7.25 (6.04–8.69) | |
| *1–3* | 15,438 | 86.89 (86.38–87.37) | | 2,246 | 85.86 (84.47–87.14) | | 6,168 | 70.85 (69.88–71.79) | | 1,020 | 68.50 (66.10–70.81) | |
| *4–6* | 1,938 | 10.91 (10.46–11.37) | | 279 | 10.67 (9.54–11.91) | | 1,592 | 18.29 (17.49–19.11) | | 317 | 21.29 (19.28–23.44) | |
| *6+* | 228 | 1.28 (1.13–1.46) | | 41 | 1.57 (1.16–2.12) | | 170 | 1.95 (1.68–2.27) | | 44 | 2.96 (2.21–3.95) | |
| Age at first birth | | | <0.001 | | | <0.052 | | | <0.001 | | | <0.001 |
| *<18 years* | 2,039 | 11.48 (11.02–11.95) | | 304 | 11.62 (10.45–12.91) | | 1,754 | 20.15 (19.32–21) | | 305 | 20.48 (18.51–22.61) | |
| *18–25 years* | 13,293 | 74.81 (74.17–75.45) | | 1,971 | 75.34 (73.66–76.96) | | 6,190 | 71.1 (70.14–72.04) | | 1,058 | 71.05 (68.70–73.30) | |
| *26–30 years* | 1,958 | 11.02 (10.57–11.49) | | 274 | 10.47 (9.36–11.71) | | 622 | 7.14 (6.62–7.70) | | 99 | 6.65 (5.49–8.03) | |
| *over 30* | 478 | 2.69 (2.46–2.94) | | 67 | 2.56 (2.02–3.24) | | 140 | 1.61(1.36–1.89) | | 27 | 1.81 (1.25–2.63) | |
| Place of delivery | | | <0.001 | | | <0.001 | | | <0.001 | | | <0.001 |
| *Home* | 2,301 | 12.95 (12.46–13.45) | | 457 | 17.47 (16.06–18.97) | | 1,375 | 15.79 (15.04–16.57) | | 265 | 17.80 (15.94–19.82) | |
| *Public health center* | 7,469 | 42.04 (41.31–42.76) | | 1,027 | 39.26 (37.4–41.15) | | 2,809 | 32.27 (31.29–33.25) | | 487 | 32.71 (30.37–35.13) | |
| *Private health center* | 4,353 | 24.50 (23.87–25.14) | | 626 | 23.93 (22.33–25.60) | | 1,626 | 18.68 (17.87–19.51) | | 299 | 20.08 (18.12–22.19) | |
| *Others: NGOs* | 3,645 | 20.51 (19.93–21.11) | | 506 | 19.34 (17.87–20.90) | | 2,896 | 33.26 (32.28–34.26) | | 438 | 29.42 (27.15–31.78) | |

includes preeclampsia and gestational diabetes, which is a risk factor of spontaneous abortion [32,33]. In some cases, obesity may make diabetes harder to manage and elevate the risk of complications, especially in the first trimester of pregnancy [4].

To check the association of mother's BMI with fetal death, this study found a positive relationship between obesity and stillbirth. This finding was consistent with the results of previous studies conducted through systematic review and meta-analysis in high-income countries [34], including Sweden [1], Denmark [13,35], Finland [36] and England [27]. Another review showed that mothers with obesity and obesity with co-morbidity had a greater relative risk for stillbirth than mothers with normal weight [37]. The possible explanation for the link between a mother's BMI and stillbirth could be that obesity during pregnancy increases the risk for other co-morbidities that result in stillbirth. Possibly, women with low weight status have better sense and capacity to feel the fetal movement and could ask for immediate care as movement declined [37,38].

**Table 3. Multivariate analysis using logistic regression for fetal and neonatal death.**

| | Miscarriage | | Stillbirth | |
|---|---|---|---|---|
| | Unadjusted OR (95%CI) | Adjusted OR[1] (95%CI) | Unadjusted OR (95%CI) | Adjusted OR[1] (95%CI) |
| **Exposure variable** | | | | |
| Mothers BMI | | | | |
| *Underweight* | 0.92*** (0.88–0.95) | 0.95** (0.91–0.99) | 1.03 (0.93–1.14) | 0.96 (0.87–1.06) |
| *Healthy weight (ref)* | | | | |
| *Overweight* | 1.16*** (1.11–1.21) | 1.08*** (1.04–1.13) | 0.95 (0.86–1.06) | 1.08 (0.97–1.20) |
| *Obesity* | 1.39*** (1.32–1.47) | 1.26*** (1.20–1.33) | 1.20** (1.05–1.36) | 1.46*** (1.27–1.67) |
| | **Early neonatal mortality** | | **Late neonatal mortality** | |
| | Unadjusted OR (95%CI) | Adjusted OR[1] (95%CI) | Unadjusted OR (95%CI) | Adjusted OR[1] (95%CI) |
| **Exposure variable** | | | | |
| Mother's BMI | | | | |
| *Underweight* | 1.14*** (1.09–1.21) | 1.07** (1.01–1.13) | 1.15* (1.01–1.30) | 1.06 (0.93–1.20) |
| *Healthy weight (ref)* | | | | |
| *Overweight* | 0.89*** (0.84–0.95) | 1.02 (0.95–1.08) | 0.91 (0.79–1.04) | 1.05 (0.90–1.21) |
| *Obesity* | 0.97* (0.90–1.05) | 1.18*** (1.08–1.28) | 0.92 (0.76–1.11) | 1.13 (0.93–1.38) |

P-values:

***$P < 0.001$,

**$P < 0.01$,

* $P< 0.05$.

[1]Only exposure variables are reported in the adjusted model.

The models were adjusted with age, education, employment status, wealth index, civil status, area of residence, age at first marriage, number of children, age at first birth, and place of delivery.

The study also revealed that early neonatal death is positively associated with maternal underweight and obesity status. This result was consistent with previous studies in which an infant born to an overweight or obese woman was at higher risk of early neonatal death than that born to a woman with normal weight [11,27,37,39–41]. In developed countries such as the USA and England, the odds of neonatal deaths were two to three times higher in infants born to overweight or obese mothers compared with those born to mothers with the recommended BMI [27,39,41]. In a pooled analysis in 27 sub-Saharan African countries, maternal obesity was identified as a risk factor, particularly for neonatal death that occurs in the first two days of life [11]. Other than obesity, being underweight was also recognised as a risk factor in this study. A recent study in India also found that underweight mothers had higher risks of neonatal death than recommended weight [42]. However, other studies observed no significant association [11,27]. The possible reason for the difference in the results could be the variations in food consumption behaviour in different regions. The most probable cause of being underweight in the South and South-East regions would be malnutrition, which differs from developed countries [43]. This paper also noted no statistically significant relationship in the latter half of the neonatal period possibly because of low statistical power. This result supported the early study in sub-Saharan Africa, in which the authors mentioned similar reasons for not finding any significant association between obesity and late neonatal mortality [11].

The study enriched the current literature by using a large sample of 193,895 women aged between 15 and 49 years old and their offspring in eight South and South-East Asian countries. To our knowledge, this research is among the initial attempts that comprehensively summarised the association between maternal BMI and fetal and neonatal death. Hence, this study has advanced the existing knowledge base by considering several subtypes of outcomes,

**Table 4. Multivariate multilevel logistic regression for fetal and neonatal death.**

| | Miscarriage | | Stillbirth | |
| --- | --- | --- | --- | --- |
| | Model 1 | Model 2 | Model 1 | Model 2 |
| **Exposure variable** | | | | |
| Mother's BMI | | | | |
| *Underweight* | | 0.97 (0.93–1.01) | | 0.97 (0.88–1.07) |
| *Healthy weight (ref)* | | | | |
| *Overweight* | | 1.06** (1.02–1.11) | | 1.03 (0.93–1.15) |
| *Obesity* | | 1.19*** (1.13–1.26) | | 1.31*** (1.14–1.51) |
| **Intercept** | 0.11*** (0.07–0.19) | 0.07*** (0.04–0.12) | 0.01*** (0.01–0.02) | 0.01*** (0.01–0.02) |
| **Model summary (Random Effect)** | | | | |
| Country Variance | 0.72 (0.44–1.18) | 0.70 (0.43–1.15) | 1.01 (0.60–1.69) | 0.99 (0.58–1.66) |
| LR test | P <0.001 | P <0.001 | P <0.001 | P <0.001 |
| ICC (%) | 13.59% | 12.92% | 23.73% | 22.95% |
| Log likelihood | -58738.35 | -58201.658 | -13688.088 | -13464.331 |
| AIC | 117480.7 | 116459.3 | 27380.18 | 26984.66 |
| | **Early neonatal mortality** | | **Late neonatal mortality** | |
| | Model 1 | Model 2 | Model 1 | Model 2 |
| **Exposure variable** | | | | |
| Mother's BMI | | | | |
| *Underweight* | | 1.07* (1.01–1.13) | | 1.05 (0.93–1.19) |
| *Healthy weight (ref)* | | | | |
| *Overweight* | | 1.02 (0.96–1.09) | | 1.04 (0.90–1.21) |
| *Obesity* | | 1.20*** (1.10–1.31) | | 1.13 (0.92–1.39) |
| **Intercept** | 0.04*** (0.03–0.05) | 0.01*** (0.01–0.02) | 0.01*** (0.01–0.02) | 0.01*** (0.01–0.02) |
| **Model summary (Random Effect)** | | | | |
| Country Variance | 0.39 (0.23–0.57) | 0.37 (0.21–0.64) | 0.57 (0.31–1.05) | 0.50 (0.26–0.96) |
| LR test | P <0.001 | P <0.001 | P <0.001 | P <0.001 |
| ICC (%) | 4.42% | 3.96% | 9.06% | 6.98% |
| Log likelihood | -35471.749 | -32027.656 | -8718.8771 | -8278.1581 |
| AIC | 70947.5 | 64111.31 | 17441.75 | 16612.32 |

P-values:

***P < 0.001,

**P < 0.01,

* P< 0.05.

Model 1: No covariates controlled for.

Model 2: Only exposure variables are reported. The models were adjusted with age, education, employment status, wealth index, civil status, area of residence, age at first marriage, number of children, age at first birth, and place of delivery.

including miscarriage, stillbirth, early neonatal death and late neonatal death in the context of emerging economies. Our findings are suggestive because the study considered the subsequent health of women and offspring health. Moreover, the consequences of poor maternal health stock have been pointed out.

The findings of this study reconfirm the necessity of addressing the problems associated with obesity of women from the perspective of offspring health. Although obesity is a growing concern in terms of fetal and neonatal death, no national or central level health program has been developed to address this burden. This study's findings can be generalised to women in this region. Results form this study will serve as helpful evidence for health policymakers to

create central-level health and nutritional interventions that can prevent obesity and improve reproductive outcomes. Such interventions could be altering women's lifestyles, providing educational modules on physical activity and establish gym and activity centres, where women can perform physical exercises to modify the obesogenic environment and lifestyle choices, thereby reducing obesity.

This study's key strength is the large population-based dataset that provided sufficient power to assess the association between maternal BMI on their children's health outcomes at different stages. A large pooled dataset also helped justify prior findings in a single or group of countries in a specific region. Moreover, the study applied statistical adjustment and was able to identify the significance of maternal obesity on children's health outcomes for informed policy formulation. The limitations of the study are as follows. First, due to the cross-sectional nature of the data, this study could not establish any causal effects. Second, although this pooled analysis included large cohorts of mothers, possible heterogeneity among studies might have restricted the reliability of the result. Hence, any extrapolation for other countries required careful consideration. Finally, pregnancy complications, such as miscarriage, stillbirth and neonatal death, might share other underlying cause(s), i.e., biological conditions or unmeasured risk factors. Thus, interpretation of the findings must be performed with care.

## Conclusion

In conclusion, this study reveals that even a modest increase in maternal BMI status is associated with fetal and neonatal death after analysing an extensive national demographic survey of eight South and South-East Asian Countries. Maternal obesity affects women, but it impacts child health in the form of fetal and neonatal death. An effective intervention to maintain recommended weight and lifestyle choices of a mother is needed to reduce the number of cases of stillbirth, miscarriage and neonatal death. Besides, obese women should deliver at a centre offering services by specialist obstetrics to ensure early neonatal care. Cost-effective community-based strategies, including nutrition programs for pregnant women, development of volunteers for immediate health support, and maternal education, are also required in this region to achieve SDG Target-3, which aims to ensure healthy lives and promote people's well-being ages by 2030.

## Acknowledgments

We gratefully acknowledge measure DHS for their permission to use the Demographic Health Surveys. We are also grateful to the editors and the reviewers who provided constructive comments to help shape this paper.

## Author Contributions

**Conceptualization:** Rezwanul Haque, Syed Afroz Keramat.

**Data curation:** Rezwanul Haque.

**Formal analysis:** Rezwanul Haque, Syed Afroz Keramat, Maimun Ur Rashid Mustafa.

**Investigation:** Rezwanul Haque.

**Methodology:** Rezwanul Haque, Syed Afroz Keramat.

**Resources:** Maimun Ur Rashid Mustafa.

**Supervision:** Khorshed Alam.

**Writing – original draft:** Rezwanul Haque, Syed Mahbubur Rahman, Maimun Ur Rashid Mustafa, Khorshed Alam.

**Writing – review & editing:** Rezwanul Haque, Syed Afroz Keramat, Syed Mahbubur Rahman, Maimun Ur Rashid Mustafa, Khorshed Alam.

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
