## [Decision Letter · Decision Letter 0]

10 May 2021

PONE-D-21-09227

Association of maternal obesity with adverse pregnancy outcome and neonatal death: evidence from South and South-East Asian countries

PLOS ONE

Dear Dr. Haque,

Thank you for submitting your manuscript to PLOS ONE. After careful consideration, we feel that it has merit but does not fully meet PLOS ONE’s publication criteria as it currently stands. Therefore, we invite you to submit a revised version of the manuscript that addresses the points raised during the review process.

In particular, data analysis seems unsatisfactory.  

We look forward to receiving your revised manuscript.

Kind regards,

Calistus Wilunda, DrPH

Academic Editor

PLOS ONE

Additional Editor Comments (if provided):

1. Table 1 includes both descriptive and inferential (95% CIs) statistics. Given that the aim is not to estimate the prevalence of participants with different characteristics in the population, it is not useful to present the 95% CIs. It is more meaningful to show the distribution of the potential confounding variables (rows) according to the exposure variable (column). This will help in assessing for imbalances in the distribution of the confounders.

2. It is necessary to properly account for the sampling weights when pooling data across surveys. It is not clear how sampling weights were handled in this paper. This is useful reference when dealing with weights when analyzing DHS data Multilevel Modeling Using DHS Surveys: A Framework to Approximate Level-Weights [MR27] (dhsprogram.com)

3. It is also unclear whether other features of the DHS survey i.e. strata and clusters, were properly accounted for in the analysis.

4. Was multilevel analysis considered?

5. Please present both unadjusted and adjusted ORs.

6. In Table 3, it is unnecessary to present ORs for potential confounders.

7. Did you explore for any interactions?

8. In the tables, it is not easy to visually distinguish between variables their categories.

Journal Requirements:

Reviewers' comments:

Reviewer's Responses to Questions

**Comments to the Author**

1. Is the manuscript technically sound, and do the data support the conclusions?

Reviewer #1: Yes

Reviewer #2: Yes

2. Has the statistical analysis been performed appropriately and rigorously? 

Reviewer #1: I Don't Know

Reviewer #2: No

3. Have the authors made all data underlying the findings in their manuscript fully available?

Reviewer #1: Yes

Reviewer #2: Yes

4. Is the manuscript presented in an intelligible fashion and written in standard English?

Reviewer #1: Yes

Reviewer #2: No

5. Review Comments to the Author

Reviewer #1: Thank you for the opportunity to contribute to the peer review process for this research. The study aims to analyze the association between maternal obesity and adverse pregnancy outcome and neonatal death in South and South-East Asian countries. The study is interesting and provides novel data on the association in South and South-East Asian Countries. However, this paper has some points that need to be clarified:

Moderate considerations:

1) In Methodology section, authors should indicate which was the statistical package that they used for data analysis.

2) In Methodology section (line 106), the authors say “This research selected 8 out of 15 countries in South and South-East Asia..”. I suggest to the authors indicate which were the countries that they included in the study.

3) Lines 134-144, the explanation about the BMI is confusing. The authors mention the BMI categorization according to the WHO authors: “Maternal weight status was the exposure of interest, and it was ascertained through the mother’s BMI. BMI was calculated as weight in kg divided by the height in m2, and it was categorised into four groups following the WHO guidelines, namely, underweight (BMI <18.50), healthy weight (BMI 18.50 to <25.00), overweight (BMI 25.00 to <30.00) and obese (BMI ≥30) (2)”. And after that they say “Since the targeted population of this study was South and South-East Asia, we used the cut off points of BMI for Asians, which are categorised as underweight (BMI< 18.50), healthy weight (BMI 18.50 to 22.90), overweight (BMI 23 to 26.9) and obese (BMI> 27) (19)”. If the authors used the BMI for Asians categorization, it is unnecessary to indicate the WHO BMI scores. Simply naming it, I think it is enough, and it makes the text easier to understand. It would be necessary to add the WHO reference.

4) The authors indicate in the abstract the range of years from which they collected data for the study. However this information is missing in Methodology section. I suggest that the authors also indicate this information in the text (and not only in figure 1).

Minor considerations:

5) Introduction, lines 73-76. In order to unify the referenced format, please remove the year in parentheses and indicate the reference number immediately after each author.

6) In Methodology section, lines 103-105, the authors say “The study pooled the most recent survey in eight South and South-East Asian countries”. Again, in line 107 they say “This research selected 8 out of 15 countries in South and South-East Asia”. From my point of view this information is redundant. I suggest to the authors unify this information in the same sentence/paragraph.

7) Authors should use past tense in the manuscript. For example, line 108 “The remaing seven countries are excluded..” should be written “were”. Please, check it in the whole text.

8) In References section, the reference 8 is incomplete.

Reviewer #2: Summary

This study investigated whether maternal obesity is a risk factor of adverse reproductive health outcomes (miscarriage, stillbirth, early neonatal mortality and late neonatal mortality) in South and South-East Asian countries.

The study used pooled data from the most recent DHS survey from 8 South and South-East Asian countries. Multivariate logistic regression was deployed to check the relationships between

maternal obesity and fetal deaths

It is found that maternal obesity associated with a greater risk of perinatal and early neonatal deaths.

Major comments:

Overall this paper addresses the important issues. However, a number of improvements are needed. Some key points are listed below:

• In this study, authors sometime mention perinatal and neonatal death, sometime adverse reproductive health outcome, and sometime foetal death. It should be consistence. Also, what is the meaning of foetal death?

• In the first paragraph of the introduction, authors told about adults overweight and obese. The authors should give the world statistics for maternal obese.

• In line 74-76 authors identify the key reasons for neonatal death but didn’t mention the reasons “Baroni et al. (2021) investigated the neonatal mortality rates in Brazil, whereas Liu et al. (2020) and Abdul-Mumin et al. (2021) investigated the key reasons for neonatal death in China and Ghana, respectively, including preterm birth (7–9)”

• This study used recent pooled data from 8 South and South-East Asian countries but for Bangladesh this study used old dataset.

• This study used multivariable logistic regression model to explore the association between exposure and outcome. But all the outcome proportion less than 10% better to use Poisson regression. Moreover, this study used multi-country data, so there is some country level variation for each outcome. For the multi-country data better to use multi-level approach.

• This study used some upper-middle income country and some lower-middle income countries. Authors should preform sensitivity analysis.

• In the abstract conclusion, authors mention “Obese women need to follow antenatal care visits and deliver at the institutional facility centre that can offer obstetrics and early neonatal care” but they didn’t discuss in the main conclusion. Conclusion should be consistence.

• In the conclusion, authors mention “An effective intervention to maintain recommended weight and lifestyle choices of mother is needed to reduce the number of cases of stillbirth, miscarriage and neonatal death. Moreover, cost-effective community-based strategies to improve maternal education are required in this region to achieve SDG Target-3, which aims to ensure healthy lives and promote the well-being of people of all ages by 2030”. It would be better if the authors mention what kind of effective intervention and cost-effective community-based strategies are needed?

Minor comment:

• I found several mistakes in references styles. If the reference has doi and volume number don’t need to write [Internet] and available from. Please follow the journal reference style.

• Reference number 8 incomplete.

6. PLOS authors have the option to publish the peer review history of their article (what does this mean?). If published, this will include your full peer review and any attached files.

Reviewer #1: No

Reviewer #2: **Yes: **Md Rashedul Islam

---

## [Author Response · Author response to Decision Letter 0]

11 Jul 2021

Please see the attachment " Response to reviewer"

---

## [Decision Letter · Decision Letter 1]

2 Aug 2021

PONE-D-21-09227R1

Association of maternal obesity with fetal and neonatal death: evidence from South and South-East Asian countries

PLOS ONE

Dear Dr. Haque,

Thank you for submitting your revised manuscript to PLOS ONE. One of the reviewers has asked for further clarification on data analysis, particularly on how weighting of the pooled data was performed. Therefore, we invite you to submit a revised version of the manuscript that addresses the points raised during the review process.

We look forward to receiving your revised manuscript.

Kind regards,

Calistus Wilunda, DrPH

Academic Editor

PLOS ONE

Journal Requirements:

Reviewers' comments:

Reviewer's Responses to Questions

**Comments to the Author**

1. If the authors have adequately addressed your comments raised in a previous round of review and you feel that this manuscript is now acceptable for publication, you may indicate that here to bypass the “Comments to the Author” section, enter your conflict of interest statement in the “Confidential to Editor” section, and submit your "Accept" recommendation.

Reviewer #2: All comments have been addressed

2. Is the manuscript technically sound, and do the data support the conclusions?

Reviewer #2: Yes

3. Has the statistical analysis been performed appropriately and rigorously? 

Reviewer #2: Yes

4. Have the authors made all data underlying the findings in their manuscript fully available?

Reviewer #2: Yes

5. Is the manuscript presented in an intelligible fashion and written in standard English?

Reviewer #2: Yes

6. Review Comments to the Author

Reviewer #2: Thank you for addressing my comments. I am satisfied to your answer. Still, I have some quarries.

1. You have pooled data from 8 countries. Where is weighting? I mean your data weighting after pooling as you deal with multiple countries with wide variation in their population? E.g. India have 87.05% population (Figure 1). Could you please explain more detail about weight?

2. It is not clear how weighting was done in the multilevel analysis.

3. It is not clear how many levels of multilevel analysis were performed. Please mention in the statistical analysis section.

4. This study conclusion and recommendation not consist with study findings. In the conclusion this study mentions about antenatal care visit but didn’t use the variable in the analysis.

7. PLOS authors have the option to publish the peer review history of their article (what does this mean?). If published, this will include your full peer review and any attached files.

Reviewer #2: No

---

## [Author Response · Author response to Decision Letter 1]

7 Aug 2021

Please see the attachment "Response to reviewer"

---

## [Decision Letter · Decision Letter 2]

16 Aug 2021

Association of maternal obesity with fetal and neonatal death: evidence from South and South-East Asian countries

PONE-D-21-09227R2

Dear Dr. Haque,

We’re pleased to inform you that your manuscript has been judged scientifically suitable for publication and will be formally accepted for publication once it meets all outstanding technical requirements.

Kind regards,

Calistus Wilunda, DrPH

Academic Editor

PLOS ONE

Additional Editor Comments (optional):

Reviewers' comments:

Reviewer's Responses to Questions

**Comments to the Author**

1. If the authors have adequately addressed your comments raised in a previous round of review and you feel that this manuscript is now acceptable for publication, you may indicate that here to bypass the “Comments to the Author” section, enter your conflict of interest statement in the “Confidential to Editor” section, and submit your "Accept" recommendation.

Reviewer #2: All comments have been addressed

2. Is the manuscript technically sound, and do the data support the conclusions?

Reviewer #2: Yes

3. Has the statistical analysis been performed appropriately and rigorously? 

Reviewer #2: Yes

4. Have the authors made all data underlying the findings in their manuscript fully available?

Reviewer #2: Yes

5. Is the manuscript presented in an intelligible fashion and written in standard English?

Reviewer #2: Yes

6. Review Comments to the Author

Reviewer #2: (No Response)

7. PLOS authors have the option to publish the peer review history of their article (what does this mean?). If published, this will include your full peer review and any attached files.

Reviewer #2: No

---

## [Editor Report · Acceptance letter]

26 Aug 2021

PONE-D-21-09227R2 

Association of maternal obesity with fetal and neonatal death: evidence from South and South-East Asian countries 

Dear Dr. Haque:

I'm pleased to inform you that your manuscript has been deemed suitable for publication in PLOS ONE. Congratulations! Your manuscript is now with our production department. 

Kind regards, 

on behalf of

Dr. Calistus Wilunda 

Academic Editor

PLOS ONE